# Monitoring of the Dehydration Process of Apple Snacks with Visual Feature Extraction and Image Processing Techniques

Diana Baigts-Allende [1], Milena Ramírez-Rodrígues [2,*] and Roberto Rosas-Romero [3,*]

1 Faculty of Agrobiology, Czech University of Life Sciences Prague, Food and Natural Resources, Kamýcká 129, 16500 Prague, Czech Republic
2 Department of Bioengineering, Tecnologico de Monterrey, Vía Atlixcáyotl 5718, Reserva Territorial Atlixcáyotl, Puebla 72453, Puebla, Mexico
3 Department of Electrical and Computer Engineering, Universidad de las Américas Puebla, Ex Hacienda Sta. Catarina Mártir, San Andrés Cholula 72810, Puebla, Mexico
* Correspondence: milena.ramirez@tec.mx (M.R.-R.); roberto.rosas@udlap.mx (R.R.-R.)

**Abstract:** Monitoring food processing is mandatory for controlling and ensuring product quality. Most of the used techniques are destructive, arduous, and time-consuming. Non-destructive analyses are convenient for rapid and conservative food quality assessment. Color images of apple slices during the manufacturing of healthy snacks were used for monitoring the drying processing. The implementation of the image-based analysis was straightforward, feasible, and low-cost. The parameters analyzed during imagen acquisition for normalizing were: contrast enhancement, binarization, and morphologic processing, varying the illumination and reference between the positions of the camera and object under analysis. Several apple features related to color, texture, and shape were extracted with computer vision techniques and also analyzed. During image analysis, the entropy was one of the most relevant computed features according to principal component analysis, and it was also relevant in terms of physical interpretation. The average percentage of entropy increase was 19.81% in the green and blue channels, while it was 16.82% in the red channel. Other relevant visual features were the skewness and kurtosis in the RGB channels; and textural information such as contrast, correlation, and variance.

**Keywords:** fruit dehydration; process monitoring; image segmentation; color features; texture extraction; shape features

## 1. Introduction

In rural communities of developing countries, the agricultural activities related to cultivating, harvesting, and handling fruits are usually important for the region's economy; however, sometimes, these products become waste when they are not consumed or processed. The production of value-added products can represent job opportunities and new income sources for the population by introducing new feasible practices as an alternative to traditional work [1]—one of such new feasible practices is manufacturing products from fresh raw materials. For example, the dehydrated fruits which are a better choice than fast food or unhealthy and sugary snacks. Additionally, the processing required for its production does not require acquiring costly infrastructure.

The monitoring of food processing is valuable for product quality and process standardization-automation [2]. Another issue is that the outcome of food processing depends on intrinsic (raw material composition) and extrinsic (temperature, slice thickness, time) parameters [3]. Therefore, computer vision could play a significant role since vision is the primary sensor in leaving beings and artificial systems during decision-making and process assessment. Besides the potential use of computer vision for food processing, these systems are effective and low-cost. Moreover, the sensor of any computer system is a camera, which is already portable and available on multiple daily devices.

The use of computer vision during monitoring tasks has been successfully applied in different fields (health care, vigilance, and environment). For instance, in the analysis of sequences of medical images to follow the evolution of lesions or [4], and recognition and tracking of smoke within video signals for detection and monitoring of forest fires for environmental protection [5], and machine vision to address the quality assessment of emulsions in the pharmaceutical industry [6]. In addition, computer vision has been exploited in the food industry to track changes in food properties during processing. For example, computer vision (CV) has been previously used as a non-destructive method for monitoring the drying behavior of organic apple cylinders subjected to different antibrowning treatments [7]. Also, drying processing of turmeric slices has been monitored by imaging features by extracting attributes such as color, morphology, texture, browning, and shrinkage [8]. Computer vision has also been used during the salting process of ham to monitor the salt diffusion within lean and fatty tissue areas [9].

Besides human vision, other sensory properties such as the auditory, tactile, olfactory, and gustatory senses have also been artificially implemented to convert, for instance, deterioration reactions into electrical signals processed by a neural network. The design of electronic tongues and noses combines materials' physical and chemical properties as sensors for monitoring specific characteristic compounds of aroma and taste present in the gaseous, liquid, and solid materials in food processing and control quality. An electronic nose was a successful method to detect and recognize fresh and moldy apples and the different molds used for their inoculation [10]. Quality assessment of delicious royal apple fresh and contaminated using bacterial cultures was determined using an in-line electronic nose system. This study's principal component analysis and Ward's analyses showed a correlation for differentiation of fresh, half, and total contaminated apples [11]. Baietto et al. (2015) have reported the use of electronic nose devices (gas sensors) for the identification of fruit volatile organic compounds for identifying the type of fruit, ripeness, and quality [12]. Applying different data-recognition algorithms to signals detected from metal oxide semiconductor (MOS) sensors was a non-invasive and rapid method to identify apple pesticide residues [13]. Effects of four drying methods (air drying, freeze drying, freeze drying plus microwave vacuum, air drying, and explosion puffing drying) on the color, texture, sensory quality, microstructure, bacterial viability, and storage stability of probiotic-enriched apple snacks were assessed. Twenty trained panelists described the texture, flavor, color, taste, and overall acceptability of the samples based on the 10-point hedonic scale [14]. Fruit samples were dried by convective, microwave-vacuum, and a combined method. Apple slices were previously dehydrated with (1) two hours of erythritol, xylitol, and sucrose; (2) thirty minutes of ultrasound. The aim was to characterize the impact of osmotic dehydration, sonication pre-treatment, and drying method on the physicochemical properties of the dried apples [15].

In the current study, apple characteristics during the drying processing were monitored using computer vision techniques as no invasive and fast methods for food control quality. Various existing image techniques were used in this research. How these techniques are combined and integrated depends on the problem to solve. That is why the image-based methodology reported in this work was specifically planned to monitor the drying. The objective of this study was to use simple and easy-to-use devices such as a digital scale and a mobile phone to monitor weight and capture images during the apple slices drying process that could be further analyzed using computer vision techniques as an alternative non-invasive and fast method for food quality control in rural communities.

## 2. Materials and Methods

### 2.1. Apple Snacks Preparation

The apples were washed, soaked in ionic silver solution (0.00014%) for disinfection for 10 min, and cut into slices (4–5 m) using an electric slicer (Chefman RJ49, Mahwah, NJ, USA). Afterward, the slices were dehydrated using an air dryer for 14 h at 68 °C. Dried samples were let cool until room temperature, weighed (25 g), and sealed in pouch bags.

### 2.2. Computer Vision Measurements

During the drying time interval, eight apple slices from different positions (on the dryer trays) were sampled nine-time instances. The change in weight using a digital balance (OHAUS Scout, Parsippany, NJ, USA) and images for further computer vision analysis were taken during drying. The images were captured with an iPhone XR (Cupertino, CA, USA), where the resolution of each image is 3024 × 4032 pixels and processed in MATLAB 2071a (Mathworks, Natick, MA, USA) (Figure 1). The visual features extracted by computer vision during the dehydration process were color, texture, and shape.

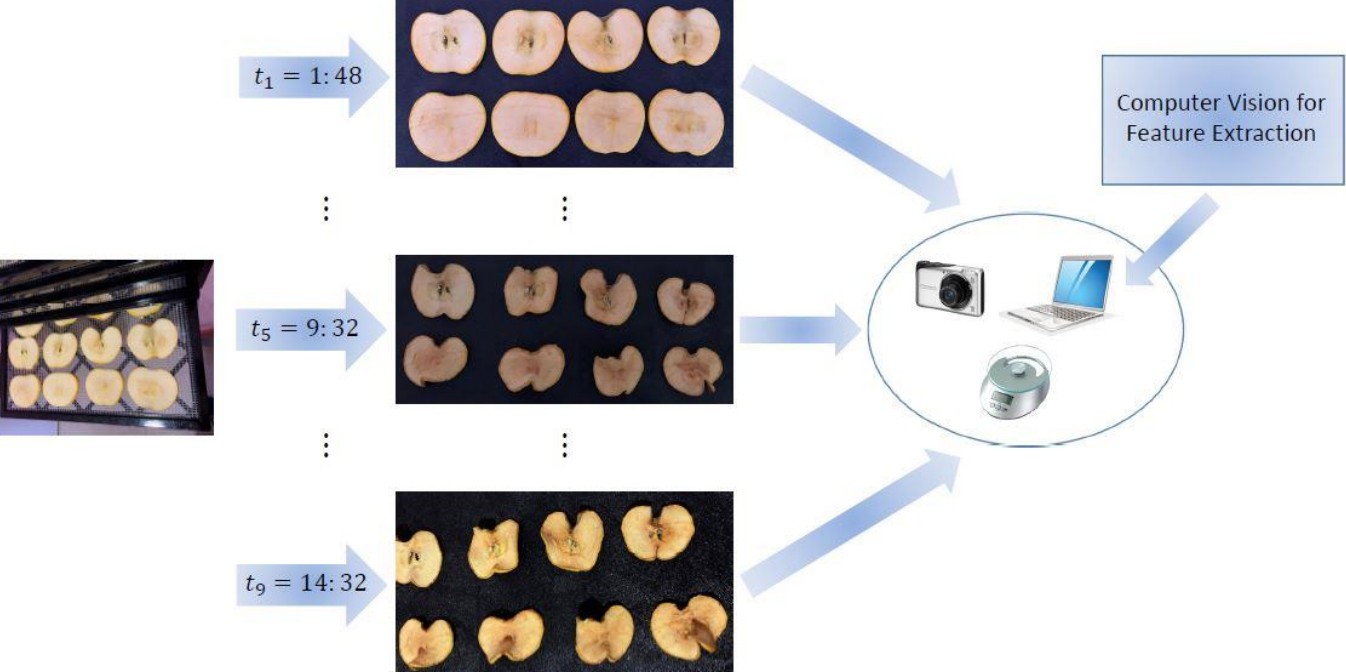

**Figure 1.** General overview of the computer vision methodology.

For this, the image is partitioned into multiple regions (Figure 2 image segmentation). A grayscale image is represented as a function on the two-dimensional space $I(r, c)$, where $r$ and $c$ specify the row and column, and $I$ is the intensity pixel value at that spatial position. A color image is obtained by mixing three-color channels: red $I_R(r, c)$, green $I_G(r, c)$, and blue $I_B(r, c)$, called RGB, and it is the standard method for the generation of color images on screens, such as computer monitors.

Panels A to E in Figure 2 show the steps for segmentation of one apple slice after 1 h and 48 min of dehydration. Panels F to J (Figure 2) show the segmentation process for the same apple slice after 9 h and 32 min. Contrast enhancement was used to adjust the range of gray intensity so that all the available intensity values (from 0 to 255) could be used, as observed in panels B and G (Figure 2), which show the result of converting a color image into a grayscale image followed by contrast enhancement.

Afterward, an image was segmented by assigning one binary value to each pixel depending on a threshold. If a pixel value is below a threshold, $I(r, c) < T$, then the output binary image is assigned 0 at that position; otherwise, it is assigned 1. This segmentation method is called image thresholding, where the threshold was computed by using the image histogram and Otsu's method. Panels C and H (Figure 2) show the result of image thresholding applied to a grayscale image. The image histogram is a function that specifies the distribution of pixel intensity values according to $P(I) = \frac{n(I)}{N}$; where $n(I)$ is the number of pixels with the intensity $I$ and $N$ is the total number of pixels within the image. Otsu's method is a statistical algorithm that determines the optimal threshold to classify pixels into foreground or background by examining the image histogram.

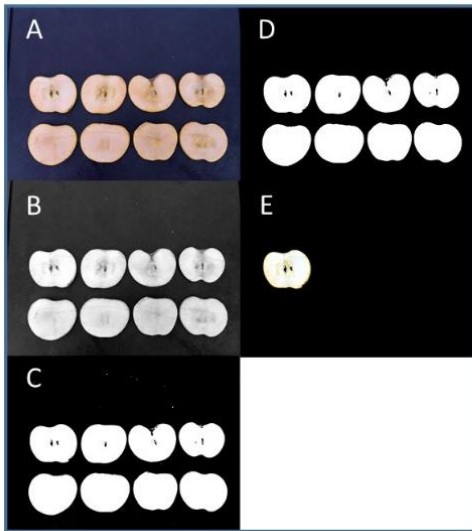 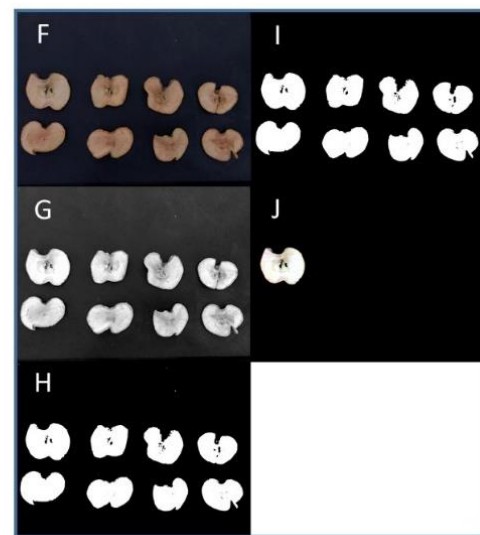

**Figure 2.** Image segmentation steps for detection of an apple slice at two time-instances: original image (**A,F**), contrast enhancement (**B,G**), thresholding (**C,H**), filtering of the largest labeled components (**D,I**), slice detection (**E,J**).

After thresholding, a segmented image contained undesired regions such as crumbs and noise. The eight largest regions, corresponding to apple slices, were kept. In contrast, the others were suppressed, as it is shown in panels D and I (Figure 2). Extraction of connected components was used to label each apple slice within the binarized images shown in panels C and H (Figure 2). Finally, each slice was detected, as shown in panels E and J (Figure 2). Features were extracted from each detected slice.

*2.3. Feature Extraction*

To obtain color features, three histograms $\{P_R, P_G, P_B\}$ were generated. One histogram was obtained from each RGB channel. Five statistical attributes were computed for each channel: the average value $\mu = \sum_{I=0}^{255} P(I)\, I$, variance $\sigma^2 = \sum_{I=0}^{255} P(I)\, (I - \mu)^2$, skewness $m_3 = \sum_{I=0}^{255} P(I)\, (I - \mu)^3$, kurtosis $m_4 = \sum_{I=0}^{255} P(I)\, (I - \mu)^4$, and entropy $H = -\sum_{I=0}^{255} P(I)\, log_2 P(I)$. Figure 3 shows the three RGB channels of an apple slice and the corresponding histograms.

For texture analysis, the color image was transformed into a grayscale image using the luminosity model, which consists of a weighted average of the RGB channels: $I = 0.21R + 0.72G + 0.07B$. According to the luminosity model, the green channel is the one that contributes the most, which agrees with the fact that green is the dominant color in apples. Textural features convey statistical information about the relative positions of the pixel intensity values within the region of interest. Textural features are obtained from the gray level co-occurrence matrix (GLCM), which specifies the distribution of pairs of pixel intensities according to the distance between these two pixels and the angle of the line segment that joins them. There are four possible angles: 0° (horizontal), 45° (diagonal), 90° (vertical), and 135° (anti-diagonal). The GLCM of a pair of pixel intensities $(I_m, I_n)$ at distance $d$ and angle $\varphi$ is defined as

$$P(I_m, I_n, d, \varphi) = \frac{Number\ of\ pairs\ (I_m,\ I_n)\ at\ distance\ d\ and\ angle\ \varphi}{Total\ number\ of\ possible\ pairs}$$

where $m, n = 1, 2, \ldots, N$, and $N$ is the number of pixel intensities. Seven textural features were computed from the GLCM: (1) the angular second moment $ASM = \sum_{m=1}^{N} \sum_{n=1}^{N} P(I_m, I_n, d, \varphi)^2$, (2) the contrast $C = \sum_{l=1}^{N} l^2 \left[ \sum_{m=1}^{N} \sum_{n=1,\ |m-n|<l}^{N} P(I_m, I_n) \right]$, (3) the inverse difference moment $IDF = \sum_{m=1}^{N} \sum_{n=1}^{N} \frac{P(I_m,\ I_n,\ d,\ \varphi)}{1+(I_m-I_n)^2}$, (4) the correlation $Corr = \frac{\sum_m \sum_n I_m\, I_n\, P(I_m,\ I_n) - \mu^2}{\sigma^2}$, (5) the co-occurrence

matrix variance $Var = \sum_{m=1}^{N} \sum_{n=1}^{N} (I_m - \mu)^2 (I_n - \mu)^2 \ P(I_m, I_n)$, (6) the difference average $DA = \sum_{m=1}^{N} \sum_{l=1}^{N} I_m \ P(I_m, I_m \pm l)$, (7) the entropy $H = \sum_{m=1}^{N} \sum_{n=1}^{N} P(I_m, I_n) \ logP(I_m, I_n)$ [16–19].

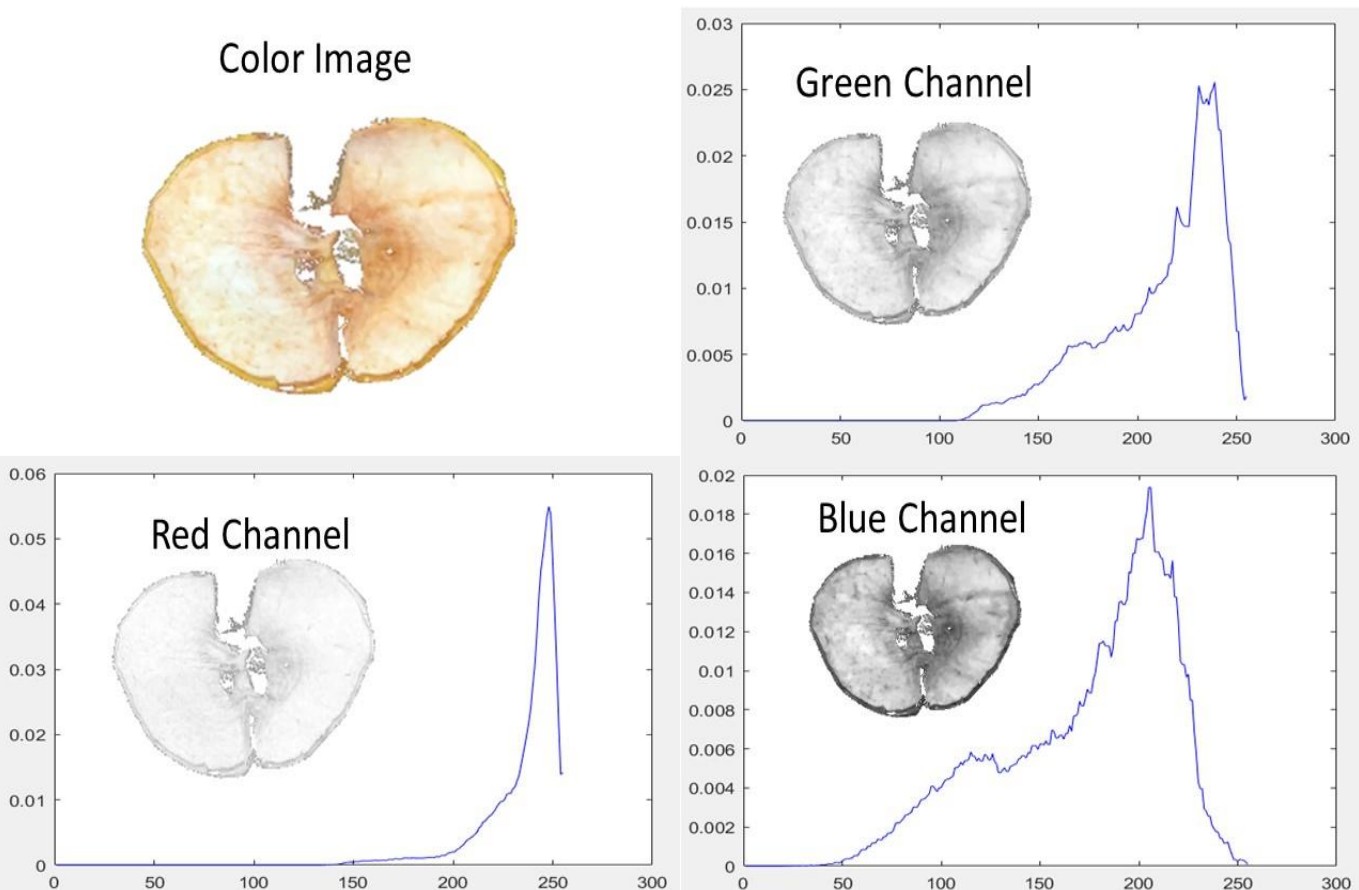

**Figure 3.** Set of histograms for the three RGB channels of an apple slice.

Shape length features were extracted from apple slices. First, the boundary contour of an apple slice was detected to obtain a sequence of straight-line segments. A straight-line segment joins two adjacent boundary pixels, and it is assigned a code number depending on its direction, as in Figure 4, where eight straight line segments are shown along with their code numbers {0, 1, 2, 3, 4, 5, 6, 7}. The sequence of segment codes obtained from the boundary contour is called chain code [20,21]. Next, an eight-entry histogram was generated from the chain code, where each entry is the number of times a code number occurs, within the chain code, divided by the chain length. This histogram contributes eight shape features, also known as direction length descriptors. Shape curvature is another shape descriptor [21], which is the frequency of occurrence of groups of external concave angles (smaller than 180°) and groups of external convex angles (larger than 180°). An external angle was formed between two adjacent straight-line segments, which were part of the boundary contour of an apple slice, scanned in the clockwise direction. Two groups of convex angles and two groups of concave angles are shown in Figure 5.

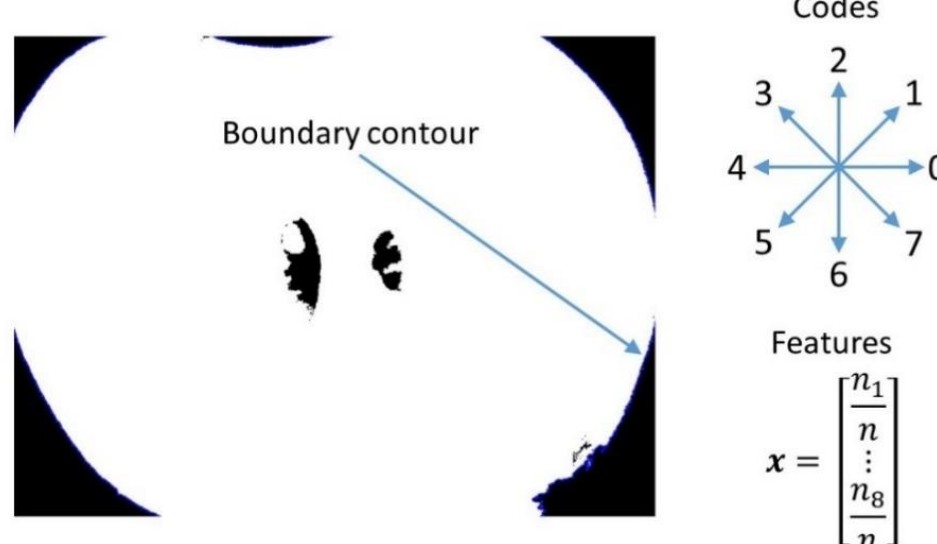

**Figure 4.** Example of the boundary contour of an apple slice, codes and directions of eight straight line segments (codes), shape feature vector.

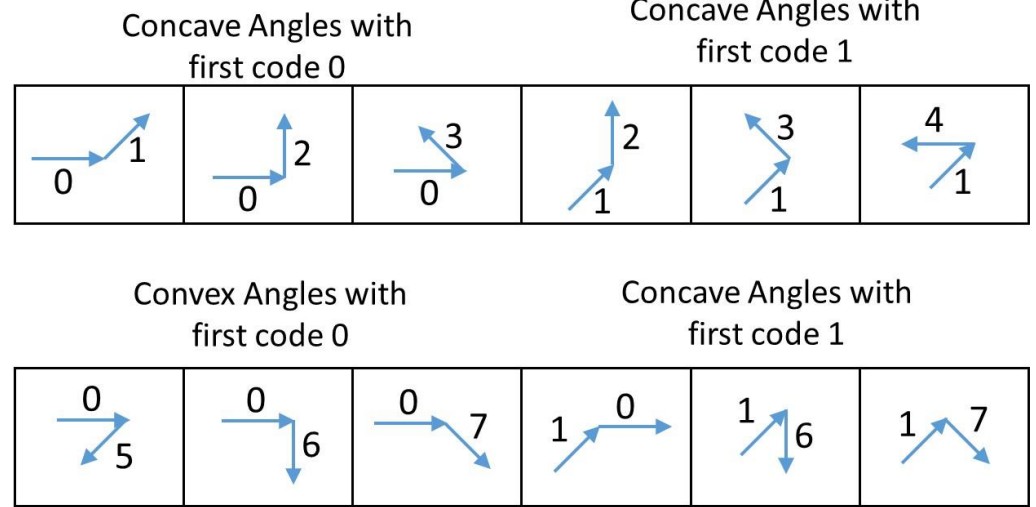

**Figure 5.** The first row contains six concave angles, and the bottom row contains six convex angles. At each row, the first three angles are characterized by the first code being zero. The last three angles are characterized by the first code being one.

The first half of the upper row is a group of three concave angles where the first code is 0. The second half of the upper row is a show of three concave angles where the first code is 1. The first half of the bottom row is a group of three convex angles where the first code is 0 while the second part is a group of three convex angles where the first code is 1. There are eight possibilities for the first code in an angle so that there are eight groups consisting of three concave angles and eight groups consisting of three convex angles. Thus, a sixteen-entry histogram was generated, where each entry is defined the number of times that members, within a group, occur. This histogram contributed with sixteen curvature descriptors.

### 2.4. Selection of Features

For those cases where the number of features is high, and there is an interest in determining the most relevant features, Principal Component Analysis (PCA) is used. PCA is a linear transformation where a set of extracted features $\{x_1, x_2, \ldots, x_p\}$ is transformed

into a set of new features $\left\{y_1,\ y_2,\ \ldots,\ y_p\right\}$, known as scores. Scores are defined in a *p*-dimensional space $y \in \mathcal{R}^p$, these scores are not correlated, and their variance determines the relevance of the scores.

The relationship between the original features ($x_j$; *j*=1, 2, ..., *p*) and the scores ($y_i$; *i*=1, 2, ..., *p*) is given by the set of linear combinations: $y_i = \sum_{j=1}^{p} \varphi_{i,j} x_j$ for *i*=1, 2, ..., *p*. The coefficient $\varphi_{i,j}$ is called loading, and it specifies the quantitative contribution of feature $x_j$ to score $y_i$. The variance/relevance of a score is specified by its corresponding eigenvalue. The higher the eigenvalue $\lambda_i$, the higher the relevance of the corresponding score $y_i$. Each entry is defined as the number of times members occur within a group. This histogram contributed sixteen curvature descriptors.

## 3. Results

*Apple Snacks Characterization*

Moisture content of raw apples was 83.5%, and after dehydration process the final values were around 16.5%, determined by weight differences before and after drying the samples (Table 1). Figure 6 shows the number assigned to each apple slice.

**Table 1.** Measurements of moisture attributes for the first apple slice during the dehydration process at nine instances of time.

| t (Hours) | Weight (Grams) | Solid Weight (Grams) |
|---|---|---|
| 0 | 17.58 | 2.9 |
| 1.68 | 11.84 | 2.9 |
| 4.15 | 4.6 | 2.9 |
| 6.02 | 3.23 | 2.9 |
| 8.40 | 3.12 | 2.9 |
| 9.53 | 3.1 | 2.9 |
| 10.7 | 3.08 | 2.9 |
| 11.52 | 3.08 | 2.9 |
| 12.95 | 3.07 | 2.9 |
| 14.32 | 3.06 | 2.9 |

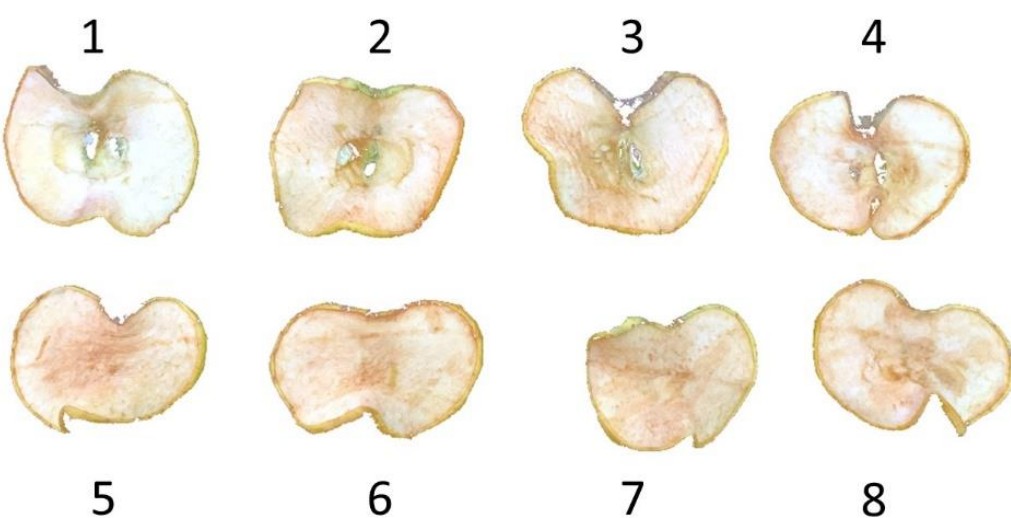

**Figure 6.** Number assigned to each apple slice. The assignation was based on its position in the oven.

It is observed that the main loss of water occurred after the first four hours, and the water removal decreased sharply after 10 h of drying. Besides weight and moisture, visual features were measured by analyzing color images in MATLAB 2071a. The images were captured with an iPhone XR (Cupertino, CA, USA), where the resolution of each image is 3024 × 4032 pixels. Initially, the apple slices were visually smooth in color and texture, and

rounded in shape. Because of moisture reduction, visual features changed from uniform to irregular. Four visual features were extracted from each apple slice: fifteen color features (five features from each RGB plane), seven texture features, eight shape length features, and sixteen curvature shape features. The 46 visual features were ranked in terms of relevance by using Principal Component Analysis. The twelve most relevant visual features are (1) entropy of the green channel, (2) entropy of the red channel, (3) skewness of the blue channel, (4) skewness of the green channel, (5) skewness of the red channel,(6) kurtosis of the blue channel, (7) entropy of the blue channel, (8) variance of the co-occurrence matrix, (9) contrast, (10) kurtosis of the green channel, (11) angular second moment, (12) correlation of the co-occurrence matrix. The occurrence matrix was used to compute the numerical value of each texture attribute, and it was obtained by considering 1 pixel between two-pixel values under study and an angle of $0°$ for the straight-line segment that joins them.

The most relevant visual features corresponded to color and texture. Features from these two groups presented advantages for visual interpretation if they were compared with shape length and curvature. Color and texture are invariant to rigid and non-rigid transformations as opposed to shape length and curvature. Rigid transformations include translation, rotation, and scaling. Figure 7 shows the rotation of an apple slice (slice 2 in Figure 6) taking place from $t_7 = 11.52$ h to $t_8 = 12.95$ h and from $t_8$ to $t_9 = 14.32$ h. During the monitoring process, the translation and rotation of apple slices were needed due to the sample's position in the dryer, the imaging capture was not the same, and the camera was unfixed. Scaling was introduced because of the distance variation between the object and camera when the image was captured. The dehydration process provokes weight loss and non-rigid deformation of each apple slice. The surface area of each apple slice was reduced (shrinkage) due to diminished moisture content. The use of features invariant to rigid and non-rigid transformations is suggested since the proposed methodology should be low-cost, simple, feasible, and effective.

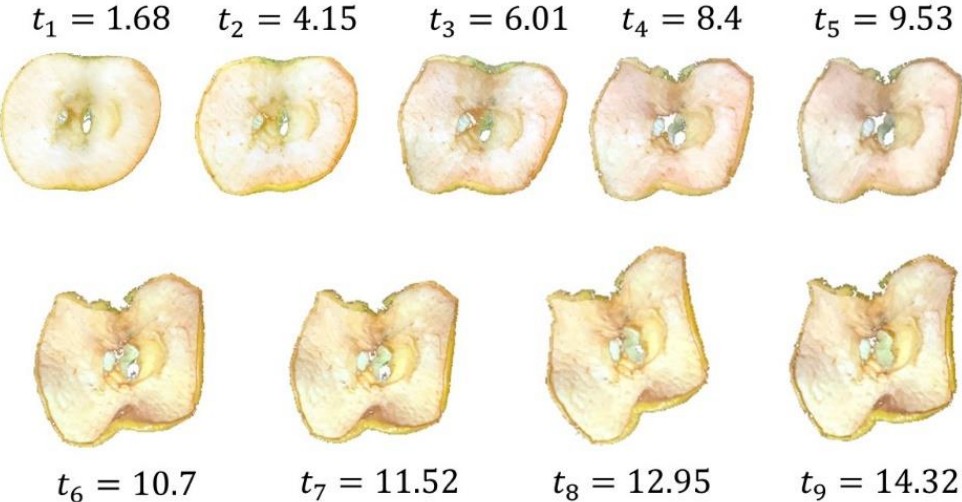

**Figure 7.** Dehydration of apple slice two over time.

Entropy was used to measure the image content in terms of visual perception. Table 2 and Figure 8 show the entropy of the green channel for eight apple slices at nine instances of time during the dehydration process. According to PCA, the entropy of the green channel was the most critical feature. Intuitively, entropy measures the random distribution of energy in a system. The temperatures of an object and the warmer surrounding equalize over time as part of the thermal energy from the warm surrounding spreads to the object. The implication that energy tends to be uniformly distributed implies maximization of entropy. As the dehydration proceeded, there was a balance between the likelihood of brownish regions (dehydrated parts) and the likelihood of yellowish regions (higher moisture). As the dehydration proceeded, the entropy trend escalated. According to

Figure 7, apple dehydration was characterized by an increase in entropy at the end of the process. Table 3 shows the average variation of the most relevant visual features during 14.32 h of the dehydration of apple slices. The average percentage of variation of the entropy over the dehydration process corresponds to an increase being 19.81% for the green channel, 16.82% for the red channel, and 19.81% for the blue channel.

**Table 2.** Entropy of the green channel of eight apple slices at nine instances of time.

| t (Hours) | 1 | 2 | 3 | 4 | 5 | 6 | 7 | 8 |
|---|---|---|---|---|---|---|---|---|
| 1.68 | 4.4659 | 5.6065 | 5.6814 | 5.8435 | 5.4219 | 5.2304 | 5.1535 | 5.4562 |
| 4.15 | 5.2655 | 5.7509 | 5.8898 | 5.7991 | 5.8423 | 5.7779 | 5.7528 | 5.9189 |
| 6.02 | 5.4095 | 6.2502 | 6.3321 | 6.2623 | 6.2990 | 6.2864 | 6.1386 | 6.2796 |
| 8.40 | 4.7413 | 5.9452 | 5.9732 | 6.0673 | 5.8624 | 5.8795 | 5.8585 | 6.1343 |
| 9.53 | 4.7388 | 5.8542 | 5.8804 | 5.9349 | 5.7763 | 5.8466 | 5.7445 | 6.0218 |
| 10.7 | 5.3979 | 6.3871 | 6.5817 | 6.4409 | 6.4760 | 6.5777 | 6.3284 | 6.7027 |
| 11.52 | 5.3229 | 6.3724 | 6.5634 | 6.4797 | 6.4882 | 6.5789 | 6.3312 | 6.6230 |
| 12.95 | 4.9871 | 6.2365 | 6.5405 | 6.4353 | 6.2559 | 6.3823 | 6.3425 | 6.5945 |
| 14.32 | 5.3671 | 6.4145 | 6.6374 | 6.5644 | 6.5484 | 6.6065 | 6.3889 | 6.7456 |

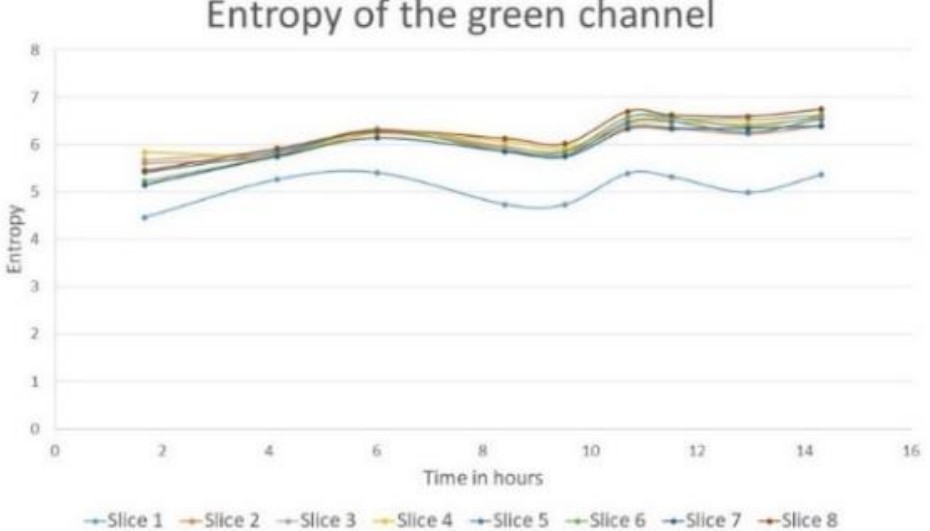

**Figure 8.** Entropy curves for the green channel of eight apple slices at nine instances of time.

**Table 3.** Average percentage of variation for visual features.

| Visual Feature | Average (%) | Visual Feature | Average (%) |
|---|---|---|---|
| Entropy green channel | 19.81 | Entropy blue channel | 19.81 |
| Entropy red channel | 16.82 | Variance co-occurrence matrix | 230.89 |
| Skewness blue channel | 11.25 | Contrast | 107.17 |
| Skewness green channel | −298.58 | Kurtosis green channel | 557.86 |
| Skewness red channel | −303.04 | Angular second moment | −37.65 |
| Kurtosis blue channel | 13.36 | Correlation co-occurrence matrix | −65.95 |

Figures 7 and 8 show that the drying curve of slice 1 (curve in blue color in Figure 8) is below the drying curves of other slices. This sample was the third heaviest slice and was in the upper right corner of the air dryer. These curves suggest that the dehydration of the first slice was faster than the other slices. Thus, the drying process was not uniform in the oven for eight apple slices.

Drying curves of other visual features (entropy of the red channel and skewness) are shown in Figures 9 and 10. Skewness measures the asymmetry of the probability

distribution of the pixel values about their mean, indicating the magnitude and direction of the deviation from the normal distribution. A negative skew usually means that the distribution tail is on the left side, while a positive skew indicates that the tail is on the right. The skewness in the green and red channels dropped while the skewness in the blue channel raised. It is observed that contrast decreased during the first four hours, then increased significantly during the next four hours, followed by a slightly decreased until the completion of drying, as shown in Figure 10. The local variation (contrast) within the slice content increased as the drying proceeded, representing higher roughness. The entropy of the red and blue channels escalated with time due to the development of brownish regions.

The box plots in Figure 11 show the variation of visual attributes over time for eight apple slices. Each box represents the time variation of a visual attribute within an apple slice. The line in the middle of a box corresponds to the median of the whole range of feature values (quartile 2). The top of the box (quartile 3) represents the median of the range of feature values between quartile 2 and the maximum, and the bottom of the box (quartile 1) represents the median of the range between the minimum and quartile 2. The cross inside a box is the average value.

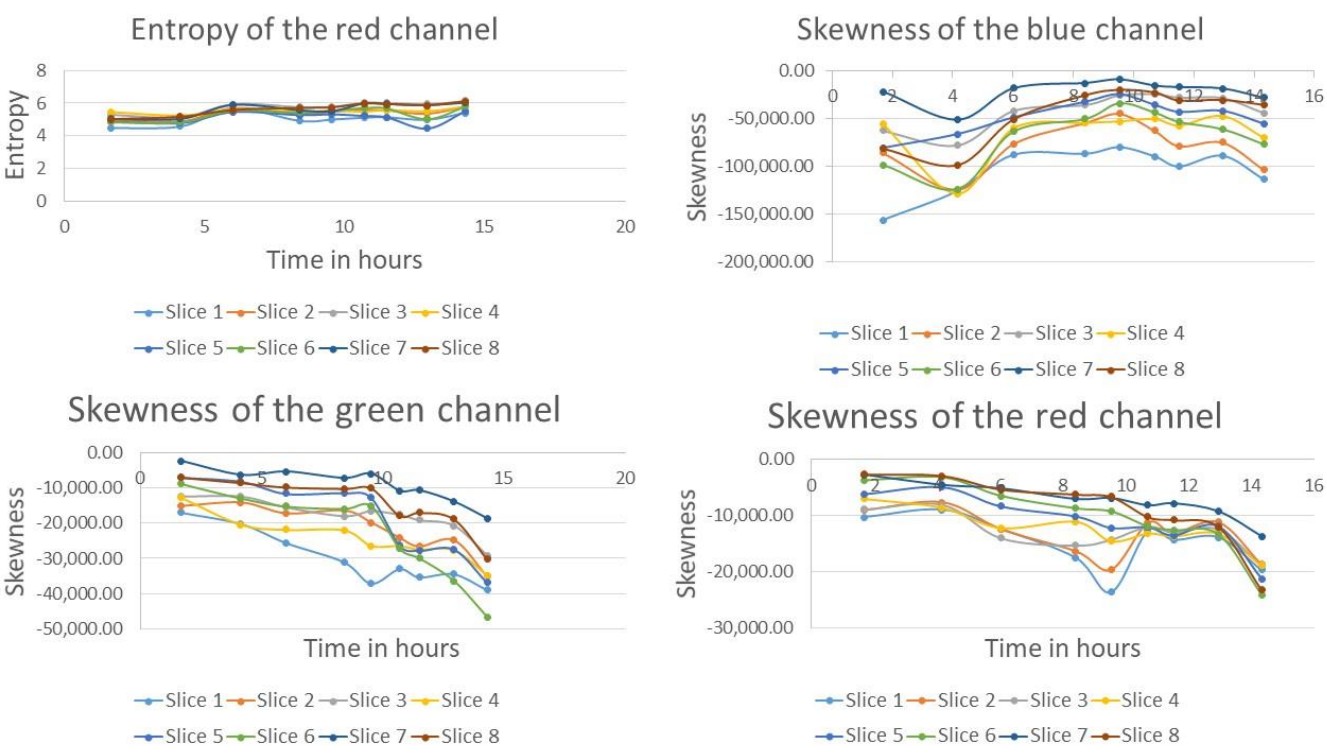

**Figure 9.** Drying curves for other visual features.

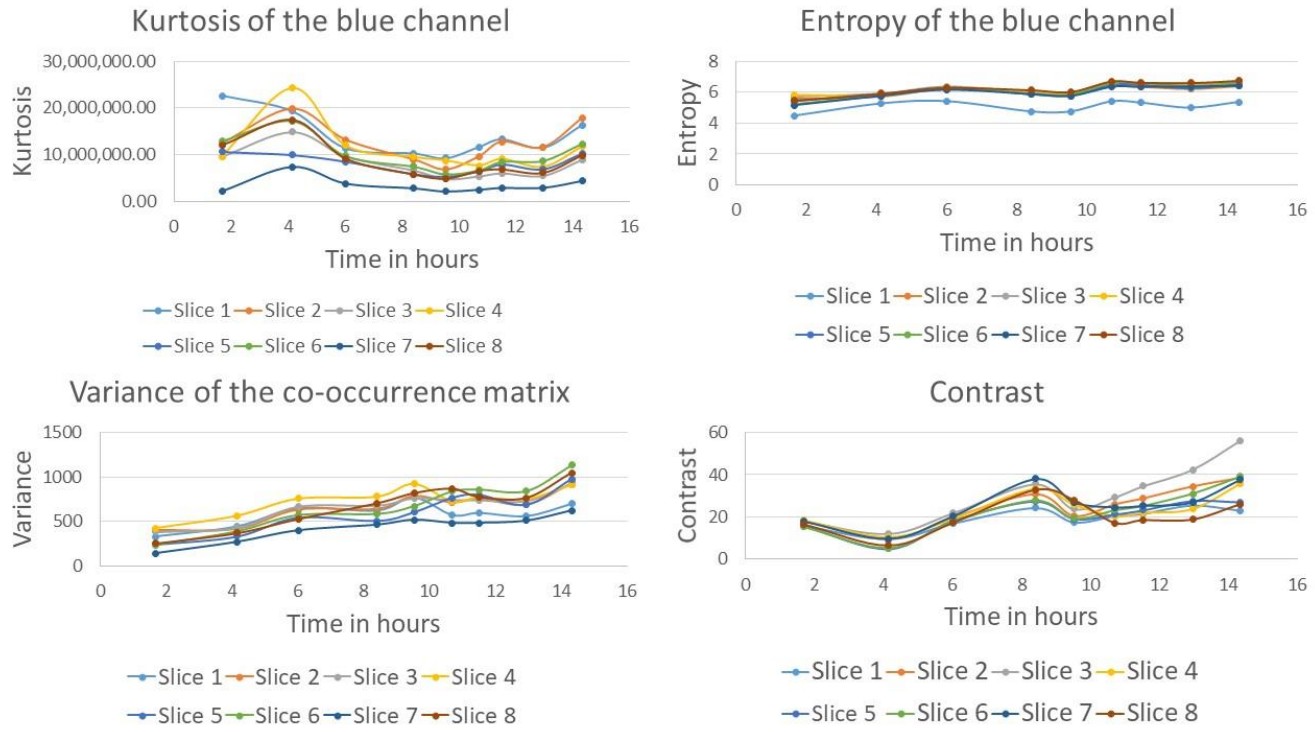

**Figure 10.** Drying curves for additional visual features.

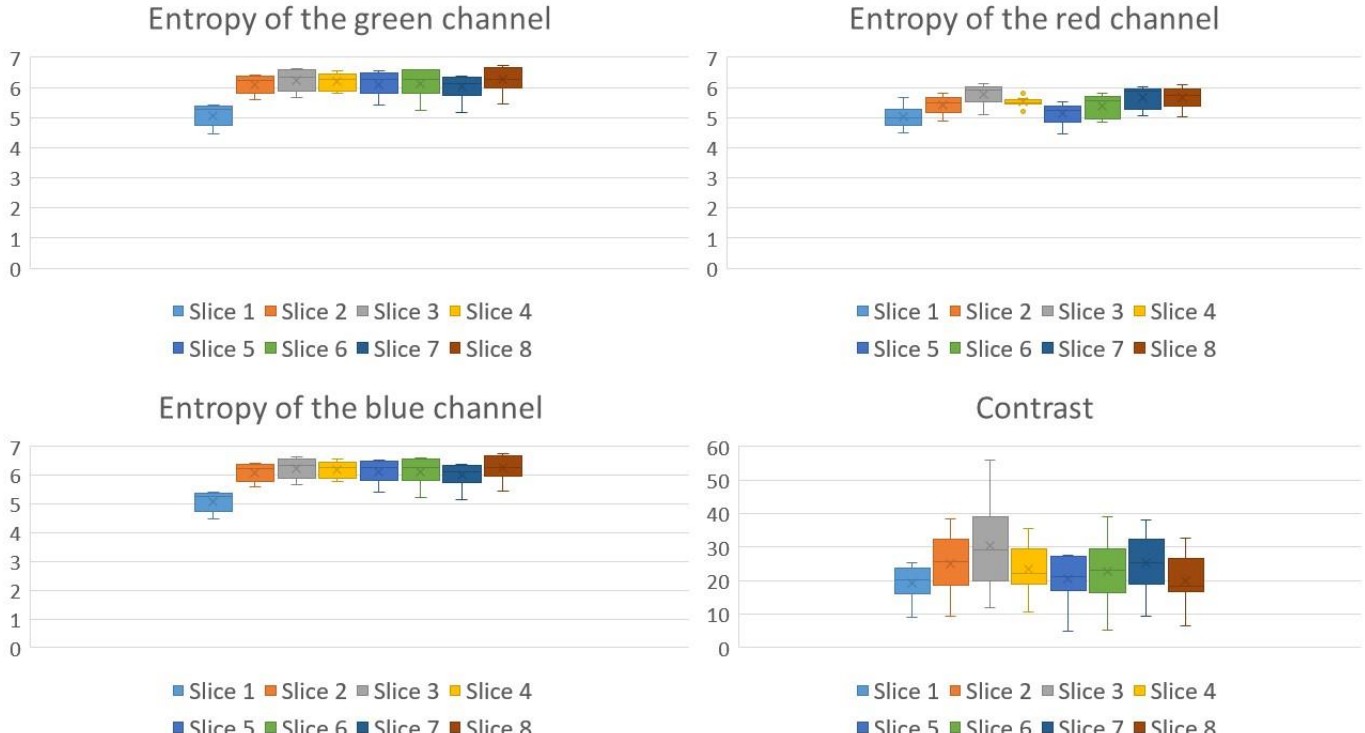

**Figure 11.** Variation of visual features of apple slices during the dehydration process: entropy of the green, red and blue channels; contrast.

## 4. Conclusions

Apple slices were dehydrated to produce healthy snacks as new labor and economic activity within a rural community. Monitoring of the dehydration process was based not only on measuring weight and moisture but also on measuring visual features such as color, texture, and forms by analyzing images captured with an iPhone. One advantage of using

computer vision to analyze the changes in food images visually is that these techniques are non-invasive and feasible to implement without using expensive laboratory equipment. The most important visual features were entropy and skew in the three RGB channels and texture contrast. Entropy was the most critical visual attribute, increasing with the uniformity of the temperature process. This characteristic could be used for monitoring changes in temperature for processing standardization.

**Author Contributions:** Conceptualization, D.B.-A. and M.R.-R.; methodology, D.B.-A., M.R.-R. and R.R.-R.; formal analysis, D.B.-A., M.R.-R. and R.R.-R.; writing—original draft preparation, D.B.-A., M.R.-R. and R.R.-R.; writing—review and editing, D.B.-A., M.R.-R. and R.R.-R.; funding acquisition, D.B.-A. All authors have read and agreed to the published version of the manuscript.

**Funding:** The Mexican National Council of Science and Technology (CONACYT) funded this research, under project grant Redes Horizontales del Conocimiento 314456 and European Union's Horizon 2020 Research and Innovation. Program under grant agreement No. 952594 (ERA Chair project DRIFT-FOOD).

**Informed Consent Statement:** Not applicable.

**Data Availability Statement:** Not applicable.

**Acknowledgments:** We acknowledge Universidad de las Américas Puebla for the infrastructure and equipment provided and the Food Analysis Laboratory, Intema S.A. de C.V. for their contribution in the performed analysis.

**Conflicts of Interest:** The authors declare no conflict of interest.

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
