# Peer review of "Monitoring of the Dehydration Process of Apple Snacks with Visual Feature Extraction and Image Processing Techniques"

_applsci, doi:10.3390/app122111269_

Round 1

Reviewer 1 Report

Color images of apple slices during the manufacturing of the apple snacks were used for monitoring the drying processing. The parameters analyzed during image acquisition for normalizing were: contrast enhancement, binarization, and morphologic processing, varying the illumination and reference between the positions of the camera and object under analysis. Several apple features related to color, texture, and shape were extracted with computer vision techniques and also analyzed. During image analysis, the entropy was one of the most relevant computed features according to principal component analysis, and it was also relevant in terms of physical interpretation. Other relevant visual features were the skewness and kurtosis and textural information such as contrast, correlation, and variance.

The implementation of the image-based analysis was straightforward, feasible, non-invasive and low-cost. The manuscript is interesting, but it needs major revision, since it is not clear what was done for the first time in present research. Has the image-based analysis been used before and for what samples? Are there any modifications of the image-based analysis with respect to previous reported data or the analysis was done in present research for the first time? Particular more detailed remarks are written further.

Particular remarks

The last paragraph (lines 64-80) of the introduction is reporting other sensory properties such as the auditory, tactile, olfactory, and gustatory senses, the design of electronic tongues and others. It is reporting pork loin, minced meat, cure ham that are not in focus of the present research dealing with apple snacks. This paragraph should be shortened and more focused to the topic. Are there any research papers dealing with he dehydration process of apple snacks? Just very simple search (even in Google scholar) is reveling the reports on this topic such as https://doi.org/10.1016/S2095-3119(17)61742-8; https://doi.org/10.3390/molecules25051078; others. There is need to include the previous results and references on apple snacks drying in the introduction part.

However, the end of introduction part is missing the paragraph about the novelty of present research. What was done for the first time? In addition, there is need to state the research hypothesis and how it will be tested? What methods will be used?

Line 84: Why the apples were washed using iodine solution?

Lines 85-86: How drying parameters were determined?

Author Response

responses reviewer 1

Reviewer 2 Report

A visual feature extraction and image processing techniques was constructed for monitoring of the dehydration process of apple snacks. Here gives some suggestions for improvement of the manuscript:

1. Introduction section, please combine the “paragraphs 1-3” into one paragraph. A final paragraph of Introduction should be added to summarize what the study did, how it was done.

2. All formulas in this article need to be reconfirmed.

3. Pay attention to the format of writing (line 73-74).

4. Repeat measurement times of this study should be added.

5. In Table 1, please explain why these times were chosen for measurement.

6. The quality of figures need to be improved.

7. Did the author use other assistive devices when collecting images with the smart device? For example, a cassette or a light source with a certain brightness.

Author Response

Responses

Round 2

Reviewer 1 Report

The authors revised the manuscript and provided explanations to the remarks.

Reviewer 2 Report

The present manuscript have gained significant improvement after revision.